# Gambling Disorder among Porto’s University Students

**DOI:** 10.3390/healthcare11182527

**Published:** 2023-09-13

**Authors:** Nádia Pais Azevedo, Paulo Santos, Luísa Sá

**Affiliations:** 1Department of Community Medicine, Information and Health Decision Sciences (MEDCIDS), Faculty of Medicine, University of Porto, 4200-393 Porto, Portugal; psantosdr@med.up.pt (P.S.); luisa.de.sa@gmail.com (L.S.); 2Center for Health Technology and Services Research (CINTESIS@RISE), Department of Community Medicine, Information and Health Decision Sciences (MEDCIDS), Faculty of Medicine, University of Porto, 4200-393 Porto, Portugal; 3Nova Via Health Unit, ACesS Espinho-Gaia, 4405-535 Valadares, Portugal

**Keywords:** gambling, pathological gambling, young adult, primary health care

## Abstract

Background: Gambling disorder is an emerging problem among young adults and must be researched to provide the necessary support. This study aims to characterise gambling disorders in Porto’s university students. Methods: A cross-sectional study distributed an online questionnaire to Porto’s university students. The authors developed a self-administered questionnaire that included the South Oaks Gambling Screen questionnaire—Portuguese Version (SOGS-PV). Results: A total of 1123 responses were included. The participants’ average age was 22.4 years (SD = 6.2), and 60.9% were women. Gambling activities were performed by 66.4% of the students, most commonly online or video games, “scratch card” games, skill games, lotteries, and sports bets. The final scores of the SOGS-PV suggested 19.7% (95% CI: 17.4–22.0) of students may have a gambling disorder, with 16.6% (95% CI: 14.4–18.8) being “probable pathological gamblers” and 3.1% (95% CI: 2.1–4.1) being “problem gamblers”. Gambling in the stocks/commodities market/virtual coins, sports bets, playing cards for money, and the numbers or betting on lotteries presents a higher risk of gambling disorder. The age (OR: 0.953; 95% CI: 0.922–0.986), being male (OR: 2.756; 95% CI: 1.899–4.000), the highest daily gambling amount (OR: 3.938; 95% CI: 2.580–6.012), the effects of the COVID-19 pandemic (OR: 0.129; 95% CI: 0.087–0.191), a mother with gambling disorder (OR: 5.284; 95% CI: 1.038–26.902), the personal services education area (OR: 2.858; 95% CI: 1.179–6.929), and the linguistics education area (OR: 2.984; 95% CI: 1.538–5.788) stand out as contributing factors to the development of this disorder. Conclusions: This study reveals a high level of possible gambling disorder among university students and emphasises the importance of this problem in the academic community. Physician awareness and prevention programmes are needed in this population.

## 1. Introduction

Gambling disorder is a non-substance-related illness characterised by persistent and recurring behaviour that causes clinically significant distress or impairment. The fifth edition of the Diagnostic and Statistical Manual of Mental Illnesses (DSM-5) points to nine criteria for the diagnosis that cannot be explained by a manic episode, of which at least four must be present over 12 months [1]. They include the need to bet ever-increasing amounts of money to achieve the desired arousal, restlessness or irritability when attempting to reduce or stop gambling, the existence of repeated, unsuccessful attempts to control, reduce, or stop gambling, frequent preoccupation with gambling, gambling when feeling distressed, the compensation behaviour of returning another day to chase the losses of money in gambling, lying to conceal the extent of gambling involvement, the impact of harming or losing a significant relationship, job, educational, or professional opportunity as a result of gambling, and relying on others for money to get out of desperate financial situations brought on by gambling [1].

According to the World Health Organization (WHO), gambling disorder prevalence in adults varies between 0.1% and 5.8% [2,3]. In Portugal, the prevalence of gambling activity in people aged between 15 and 74 years is estimated at 48%, with men being more likely to gamble than women (51.0% vs. 45.2%). Younger people (15–34 years old) present a lower prevalence (42.8%). According to the South Oaks Gambling Screen (SOGS), 1.2% of the population may have a gambling disorder, while 0.6% are likely to be pathological players [4].

Being male, single, belonging to a low socioeconomic class, having poor physical or mental health, and substance abuse, specifically alcohol and tobacco, appear to be risk factors for gambling disorder [1,5,6,7,8,9,10,11,12,13,14].

Gambling is an addictive behaviour with physical and mental health consequences that are frequently undiagnosed and untreated. It is influenced by contextual and non-contextual factors, such as individual biological and psychological differences. It appears to be linked to depression, bipolar disorder, and substance abuse, particularly alcoholism [1,2,15].

Dealing with internal states of restlessness, experiencing solid and exciting sensations, and the desire/illusion to change one’s socioeconomic status easily and quickly are some causes of gambling [6,8].

People with this disorder know that their behaviour jeopardises, destroys, and affects personal, family, and work relationships. This can be due to the frequent lies to conceal the problem, the requests for money, or the individuals’ beliefs that their earnings are related to their abilities and losses to bad luck, among other attitudes. The more they gamble, the more they believe in their skills and gains, creating a vicious cycle [1,6,16].

Furthermore, there are long-term effects on those who no longer gamble. This disorder can lead to financial distress, mental and physical health problems, strained relationships, potential legal issues, substance abuse, and family disruption, and it can affect future generations [7].

While some individuals experience the onset of gambling disorder during adolescence or youth, others may encounter it in mid-adulthood or even later in life. Notably, early signs of this problem are more prevalent among men than women. Those who initiate gambling during their youth often do so in the presence of family members. Numerous international studies consistently indicate that gambling is a prevalent aspect of young people’s lives. Despite adolescent gambling being an illicit activity, young individuals participate in gambling at a higher prevalence rate than adults [1,17].

University students face several changes in their lives in a short period of time, accompanied by a multitude of stressors inherent in academic pursuits, rendering them susceptible to the premature onset of a gambling disorder, especially if associated with impulsive traits and substance abuse [2,3,5,7,17]. Furthermore, the probability of recognising the problem and seeking treatment is very low compared to the general population [1]. Moreover, the current youth generation has grown up in an era where gambling opportunities are readily available and widespread. The normalisation of gambling in numerous countries, its extensive availability, and ease of access, along with governments’ dependency on gambling revenues, may pose a threat to achieving sustainable development goals, which demonstrates the need for more studies on this somehow neglected and understudied issue [2,4,17].

Thus, the description of this disorder and its consequences emphasises the importance of tracking this problem and providing the necessary support and responses in primary care [2,5,7]. We aim to characterise the gambling disorder in the population of young university adults and investigate its determinants to find better strategies for a preventive and therapeutic approach to this problem.

## 2. Materials and Methods

### 2.1. Study Design

A quantitative cross-sectional study was carried out based on a self-completed online questionnaire.

### 2.2. Participants

All Porto’s higher education students in public institutions, including undergraduate, master’s, doctoral, and postgraduate students, were eligible. The University of Porto (UP) has around 30,000 students, and Porto Polytechnic Institute has approximately 20,000 students. A convenience sample was used, with an estimated sample size of 380 responses.

### 2.3. Measures and Covariates

We developed a self-administered questionnaire that included academic characterisation (scholarship status, employed students, educational institution, year of study), self-perceived physical and mental health status, psychoactive substance consumption, the impact of the COVID-19 pandemic, and the South Oaks Gambling Screen, validated for the Portuguese population (SOGS-PV).

The SOGS-PV questionnaire assessed for gambling disorder. This scale has a high internal consistency, with a Cronbach’s alpha coefficient of 0.97 in the English version [5]. Kuder-Richardson (KR-20) was used in the Portuguese version and yielded a value of 0.71 [3].

Although SOGS is based on the Diagnostic and Statistical Manual of Mental Disorders, Third Edition (DSM-III) criteria, which have not yet classified gambling disorder as an independent disorder, it remains one of the most widely used in the literature and is regarded as one of the best screening tools for gambling disorder [3,18,19,20]. Furthermore, it is available in the electronic clinical process of primary health care in Portugal to be used as a screening method by family doctors.

It consists of twenty-six questions based on DSM-III pathological gambling criteria. It is suitable for screening the general population for a gambling disorder and can be self-administered or performed by non-professional or professional interviewers [5].

Most questions on this scale are answered “yes” or “no”. Six questions that characterise the various game types are not included in the final score because they do not correspond to the criteria defined by the international classifications. The total score is calculated by adding the scores from the remaining twenty items. Individuals with a score of 5 or higher are considered “problem gamblers (PG)”, those with 1 to 4 points are “probable pathological gamblers (PPG)”, and students with 0 points do not have a gambling disorder [3,5,19]. Because the purpose of this study is to emphasise the screening of gambling disorders, the participants were classified as those who do not have a gambling disorder (0 points, so it is improbable that the DSM-5 criteria apply to these individuals) and those who are problem/probable pathological gamblers (a score of 1 or greater and, thus, require further evaluation) [3,5,18,19].

### 2.4. Procedures

The invitation for participation was distributed on the University of Porto’s internal platform (*inqueritos.up.pt*) through the official Department of Communication and Image in November 2022. It was delivered at two different times, separated by one month. It was accessible for 40 days between November 2022 and January 2023. It was only possible to access with an institutional email; participants had to provide informed consent before filling it out, and it could only be submitted once. Except for open-ended questions, the questionnaire was considered complete if all questions were answered.

The participants’ anonymity was preserved throughout the entire study. The use of the UP’s internal platform guaranteed the security of the answers. The data were only available to the researcher and the supervisor on a laptop computer and an external disk that was password-protected.

### 2.5. Data Analysis

Data from the UP’s survey platform were exported to Microsoft Excel 2016, and statistical analysis was performed using IBM SPSS Statistics software (Statistical Package for Social Sciences), version 27.0.

Histograms were used to check for normality. Means and standard deviations (SD) were used to describe quantitative variables with a normal distribution, and medians and interquartile ranges were used to describe variables that did not have a normal distribution. Absolute and relative frequencies were used for categorical variables.

A multivariate analysis was performed using binary logistic regression to check the association between the type of gambling and the risk of gambling disorder. Pearson’s chi-square test assessed the relationship between nominal categorical variables and the final SOGS classification, with *p* < 0.05 considered significant.

All variables were included in the binary forward conditional logistic regression. The logistic regression was adjusted for confounding variables such as gender and educational institution. The odds ratios (OR), confidence intervals (CI), and p-values were calculated for each variable in the logistic regression. We decided to present only the ones that showed significance in the final model.

The study was conducted respecting the Declaration of Helsinki, and the protocol was approved by the Ethics Committee of São João University Hospital Centre/Faculty of Medicine of the University of Porto (257-22).

## 3. Results

A total of 1639 participants submitted the questionnaire, but only 1123 were included in the analysis since 516 had incomplete answers.

### 3.1. Characteristics of the Population

University students’ median age was 22.4 years (SD: 6.2), with the majority being female and single. Most students attend for the first three academic years. They identified themselves as coming from mainly middle-class families. In 69.7% of cases, students had to switch accommodations when they started college, and 16.2% were employed simultaneously. Alcohol was the most consumed substance, followed by tobacco and cannabis. Regarding their health assessments, they report better physical health than mental health (Table 1).

Regarding the academic areas, the health area received the most answers (about 30%), followed by the engineering sector (17%) and the biological sciences area (15%).

### 3.2. Gambling Disorder

The final scores of the SOGS-PV show that 19.7% (95% CI: 17.4–22.0) of the university students may have a gambling disorder (Table 2).

### 3.3. Types of Gambling

The most popular types of gambling among university students are online or video games (52.9%, with 28.9% of them playing once or more per week), “scratch card” games (36.8%, with 2.0% of them playing once or more per week), games of wits (35.6%, with 1.3% of them playing once or more per week), and numbers or lottery bets (26.3%, with 1.5% of them playing once or more per week) (Table 3).

Regarding the number of games, 20.4% played just one type, 24.0% played two types, and 37.4% played three or more types of games.

Almost all types of gambling were associated with the risk of gambling disorder when played once or more per week (Table 3). The multivariate analysis, using binary logistic regression to check the association between the type of game and the risk of gambler disorder, shows that gambling in the stocks/commodities market/virtual coins presents a higher risk, followed by sports bets, playing cards for money, and playing the numbers or betting on lotteries (Figure 1).

### 3.4. Money Gambled

A total of 66.4% (n = 746) of university students have already gambled money, and half of them have spent between 1 and 100€ in one day. We emphasise that 8.2% of the students have already spent more than 100€ in one day (Figure 2).

Participants who took out loans (n = 49) to gamble or pay off gambling debts did so from relatives other than their spouse or in-laws (32.0%), from money for household management (25.2%), or by selling their own or their family’s assets (20.6%).

### 3.5. Relatives with Gambling Disorder

Regarding the question, 8.2% refer to a friend or significant other; 4.7% to their grandfather or grandmother; 4.2% to their father; 2.3% to their sibling; 0.9% to their partner; and 0.8% to their mother. In addition, 11% report another relative with a gambling disorder.

### 3.6. COVID-19 Pandemic and Gambling Habits

A total of 16.2% of participants believed that the COVID-19 pandemic and subsequent confinement had affected their gambling habits (55.5% of those had a score greater than zero).

### 3.7. Determinants Associated with Gambling Disorder

The variables identified as potential predictors of a gambling disorder were age (OR: 0.953; 95% CI: 0.922–0.986), being male (OR: 2.756; 95% CI: 1.899–4.000), the highest daily gambling amount (OR: 3.938; 95% CI: 2.580–6.012), the effects of the COVID-19 pandemic (OR: 0.129; 95% CI: 0.087–0.191), a mother with gambling disorder (OR: 5.284; 95% CI: 1.038–26.902), the personal services education area (OR: 2.858; 95% CI: 1.179–6.929), and linguistics education area (OR: 2.984; 95% CI: 1.538–5.788) (Table 4).

## 4. Discussion

Our study found that 19.7% of university students in Porto may have a gambling disorder (16.6% are probable pathological gamblers and 3.1% are probable gamblers). These findings are significantly higher than the WHO’s estimate of 5.8% for the prevalence of gambling disorders in adults [2,3] and higher than the 2016 adult Portuguese context (1.2% are probable pathological gamblers and 0.6% are probable gamblers) [4]. These estimates are concerning because they come from a younger age group with higher literacy and a predominantly middle-class background. This can indicate that this is a vulnerable group with future implications.

Compared to other students’ samples, our sample exhibited notably elevated proportions in the PPG category while maintaining a comparable balance of PG. Our goal of using SOGS as a screening instrument may explain this trend. Therefore, we designated students as PPG if their corresponding SOGS scores equaled or surpassed a threshold of 1 [21,22,23,24].

The literature suggests that gambling disorders are more prevalent in males and at a younger age due to impulsivity traits, consistent with our results. Men also seem to have more difficulty stopping gambling. Being male is associated with increased cognitive bias and an increased likelihood of emotional regulation difficulties. These participants primarily engage in online or video games, “scratch card” games, skill games, lotteries, and sports bets [4,6,8,9,11,14,16,22].

Regarding the types of games, we found that almost all are associated with gambling disorders if played once or more per week. This can be explained by gambling’s capacity to activate the brain’s reward system by releasing dopamine, a neurotransmitter associated with pleasure and reinforcement. Frequent gambling can lead to a conditioned response where individuals associate gambling with positive feelings, making them more likely to engage in it repeatedly. In our sample, the types of gambling that were related to a higher risk of gambling disorder were stocks/commodities market/virtual coins, followed by sports bets, playing cards for money, and playing numbers or betting on lotteries [1,3,14].

The rise of participation in the stock market, commodities market, and virtual coins among young adults has become a notable trend. These financial activities have garnered increased publicity. Although they can be a type of gambling, they are also associated with elements of skill and knowledge, which can lead individuals to place higher bets when confident in their abilities. Nonetheless, similar to traditional forms of gambling, these financial ventures entail uncertain outcomes, stimulating the brain’s reward system and evoking an adrenaline rush, reinforcing the desire to continue trading. For some individuals, engaging in financial market trading can serve as a coping mechanism to deal with stress and negative emotions. Monitoring market movements and executing trades temporarily diverts their attention from personal challenges or provides a sense of control over their financial circumstances. The ease of accessibility offered by online trading platforms and mobile applications has significantly contributed to the popularity of these activities and has facilitated more frequent trading, potentially increasing the risk of developing addictive behaviours. Moreover, traders may develop cognitive biases, such as overconfidence or the illusion of control, leading them to believe they can consistently predict market movements and make profitable trades. These biases can perpetuate engagement in trading, even in the face of losses [1,25,26].

In Portugal, soccer matches and tournaments hold significant cultural importance and are heavily promoted. However, the existing Portuguese law concerning advertising these matches and sports betting needs to be improved. Individuals in Portugal can engage in sports betting either in person at authorised establishments or online through diverse platforms, aligning with technological advancements and the increasing prevalence of mobile devices and betting applications. Importantly, our study’s findings corroborate that sports bets are particularly favoured among the Portuguese population, with a notable emphasis on young adults [4,27,28].

Engaging in card games for monetary stakes is associated with gambling disorder due to its similarity with other forms of gambling mentioned previously. Another factor that warrants consideration in card games is the multiplayer aspect, which has been linked to an increased risk of developing a gambling disorder [28,29].

The addictive nature of scratch cards and lotteries lacks sufficient regulation compared to other gambling products, potentially fueling the continuous growth of a multibillion-dollar industry. The absence of preventive measures leaves vulnerable individuals prone to attraction and problematic use. Notably, a non-profit state-owned organisation exclusively provides scratch cards and lotteries, ostensibly promoting responsible gambling. These findings call for heightened awareness among the public, scientific community, and regulatory bodies, emphasising the need for effective interventions [30,31].

The delineation between gambling and gaming activities has become increasingly blurred due to technological convergence. Advancements in new technologies have facilitated various innovations and changes in gambling opportunities. Over the last decade, the internet has witnessed the proliferation and sophistication of gambling products, along with the emergence of unregulated gambling using virtual currencies and “gambling-like” content such as loot boxes and social casino games. As a result, academic and regulatory interest in this domain has surged, especially among younger users who frequently engage with these technologies. Some studies suggest that participation in both gambling and video game play is prevalent, with a significant majority of video game players having been involved in gambling activities in the past year and vice versa, with a substantial proportion of gamblers reporting involvement in video game play during the same period. In our sample, we questioned students about spending money online and in videogames that take advantage of this convergence and placed both in the same category [32,33].

In our study, poorer physical health, alcohol consumption, and tobacco and cannabis usage were linked to higher SOGS-PV scores. Still, none were significant, which can be explained by a lack of self-reported substance use related to self-surveys. Individuals suffering from a gambling disorder have more health conditions and often require medical care [5]. Due to gambling’s ability to trigger the same addictive pathways as other psychoactive substances, the relationship between gambling and alcohol is already well-known [14]. Drinking alcohol may also cause the student to bet more frequently and spend more money than they otherwise would. Moreover, alcohol sales and use are encouraged at most gambling venues [5,7,12,22].

Given that gambling is viewed as a social activity and that smoking is permitted in some physical gambling facilities, several research studies link gambling with tobacco abuse. Cannabis use can also be related to a higher likelihood of a person having a gambling disorder, since numerous sources support this association [1,8,34]. Furthermore, the accessibility of this activity in cafes, petrol stations, and online facilitates its dissemination [1,7,12,35].

Given that the sample was mostly made up of single people, self-reported average socioeconomic class, and self-reported good mental health, these variables were not associated with a higher likelihood of developing a gambling disorder, unlike other studies [1,7,12,35,36]. However, the university population is particularly at risk for developing these coping mechanisms since psychological stress can be the driving force for developing a gambling disorder as a maladaptive coping mechanism for unpleasant feelings [12,14]. Suicide attempts and deaths by suicide are directly correlated with high levels of gambling disorder. Respectively, people who gamble and those with a gambling disorder are 2.4 and 2.8 times more likely to attempt suicide than non-gamblers [37], emphasising how crucial it is to recognise this issue as soon as possible and develop successful methods for helping this population [2].

The scientific community has expressed concern about transitioning from traditional to online gambling. Online gambling is seen as a high-risk activity because of its accessibility, speed, and anonymity. The closure of several gambling venues has highlighted this transformation due to limitations put in place by public health due to the COVID-19 pandemic [16]. The current literature suggests that people with gambling disorders were especially vulnerable during the pandemic [16]. Our study shows that a shift in gambling habits during the COVID-19 pandemic greatly contributed to an increased risk of developing a gambling disorder. A subgroup particularly at risk for this issue is characterised by young male individuals who already have a gambling disorder, poor mental health, and high levels of substance use, which is supported by our study [35].

In our study, the maximum amount spent gambling in a single day (higher than 1€) was associated with being a PPG. However, this can be due to the group of students who had never gambled any amount and could not have a gambling disorder. Additionally, the ones who are PPG spend more than 1€ since fewer gambling activities are below 1€. Another possibility is the rapid increase in the amounts spent because of the addictive nature of gambling. Youth is also a vulnerable stage of development, marked by a shift in the type of social status students value, with their own money and position in their group becoming increasingly important. Gambling can be used as a way to assert autonomy and independence [10].

Concerning relatives or friends with a gambling problem, only a mother who is a probable gambler was linked to a score greater than 0 in our study. According to the literature, parents culturally encourage gambling behaviour in males more than in females. A distant parental relationship is also linked to developing a gambling disorder and internet and video game addiction among adolescents [8].

Regarding the study fields, the linguistic and personal services areas were related to higher SOGS-PV scores. Different activities of socialisation prevalent among these students could elucidate this association.

Despite being non-significant, the health, economics, and engineering areas are less likely to have a gambling disorder. Concerning the health area, these students’ increased awareness of the disorder can explain this trend. We are unaware of other studies correlating different fields of study with a higher risk of gambling disorder.

Gambling is an activity that is generally well accepted by society [14]. It can be used as amusement, the answer to the search for emotion, joy, excitement, or even the desire to become wealthy [38]. However, to lessen the harm to the affected person and society, it is crucial to recognise the early signs of a gambling disorder and distinguish them from the simple act of gambling [9].

To date and to the authors’ knowledge, our study is the first to assess the Portuguese university population for gambling disorder, achieving the highest number of responses to the SOGS in the university population.

Some limitations may result from volunteer bias since this study used a non-random sample (a convenience sample) weakened by the high number of incomplete questionnaires (31.5%). Other limitations are memory bias or social desirability bias, which could lead to an overestimation or underestimation of the problem’s prevalence. Concerning the pandemic effect, due to the age of our participants, these results can be biased since some of the students enrolled in university during this time were of legal age and some were underage. Additionally, the DSM-5 currently recognises gambling disorder as a separate diagnosis. Thus, since SOGS was created based on the DSM-III criteria, it is essential to update SOGS according to the most recent criteria [1]. Although some of the literature indicates several concerns about SOGS, it remains one of the best and most widely used screening tools for gambling disorders [18]. Utilising a convenience sample could potentially impact the external validity of this study, thereby inhibiting the generalisation of its results to the entire population of university students in Porto or the extrapolation of its findings to the broader Portuguese population. Nevertheless, it is noteworthy that the considerable sample size attained, along with the measures of anonymity and confidentiality implemented, bolsters our findings’ credibility and robustness.

Due to the comparable representation of both male and female participants within our sample, the opportunity to conduct a gender-based regression analysis was precluded. Such an analysis holds the potential to unveil noteworthy distinctions between these two cohorts concerning gambling preferences and potentially elucidate underlying traits contributing to the heightened susceptibility of males towards gambling disorder. It is worth noting that similar studies have undertaken gender-based regression analyses, revealing differences in gambling patterns, the amount of money gambled, and even societal scrutiny [22].

Another limitation of our study was the convergence of the online gambling and video game categories. Although it can also be an advantage since many video games are associated with gambling, this could cause some confusion among the participants, leading to individual differences in how respondents perceived these activities, who then mentioned types of video games in the open questions. This approach can introduce an overlap between online gambling, gaming with loot boxes, and gaming in general.

In future work, it would be crucial to perform a clinical interview using the DSM-5 criteria (the gold standard for assessing a gambling disorder) with participants who scored higher than zero on the SOGS-PV questionnaire to confirm this pathology, evaluate the validity of this screening tool, and assess the correlation between gambling disorder and the types of gambling practised by participants [1,15].

Assessing a younger population gives an idea of the extent of this pathology in these age groups. It also makes it easier to implement interventions that reduce and help raise awareness of gambling disorders. This can be a primary or secondary prevention intervention, but ideally in the earliest stages of the problem. Another advantage of this study is that it identifies the factors associated with gambling disorder to better establish more effective prevention and treatment protocols.

Examples of primary interventions include public awareness campaigns, educational programmes in schools, media and advertising regulations, parental involvement and training professionals on gambling disorder screening, financial literacy programmes, online gambling protections, implementing more effective age verification systems, developing and enforcing comprehensive gambling regulations and policies that prioritise player protection, responsible gambling measures, and access to treatment resources.

In future work, it will be critical to develop new screening technologies for this rising problem, be aware of and adapt to the most recent and addictive forms of gambling and find more effective therapies. Furthermore, this study highlights the importance of future research on the implications of emerging technologies and new forms of gambling in the population and, therefore, screening methods tailored to these new types of gambling.

## 5. Conclusions

Almost one out of five Porto´s university students may have a gambling disorder. These findings are concerning since we can observe how this pathology may be underdiagnosed and understudied in the population.

Specific types of games, such as gambling in stocks/commodities market/virtual coins, sports bets, playing cards for money, and playing the numbers or betting on lotteries, present a higher risk of gambling disorder.

The maximum amount spent in a single day, the change in gambling habits after the COVID-19 pandemic, a mother with a gambling disorder, the linguistics education area, the personal services education area, age, and a predisposition in males are contributing factors for developing a gambling disorder in this population.

Even though gambling is a public health concern, the role of health professionals remains undefined since there is no process to prevent or treat this pathology. This research can draw attention to an emerging problem as well as new information and associations about gambling disorder in the Portuguese context, which can be used to justify our social responsibility and government actions, such as support to fight this disorder or restrict gambling activity.

## Figures and Tables

**Figure 1 healthcare-11-02527-f001:**
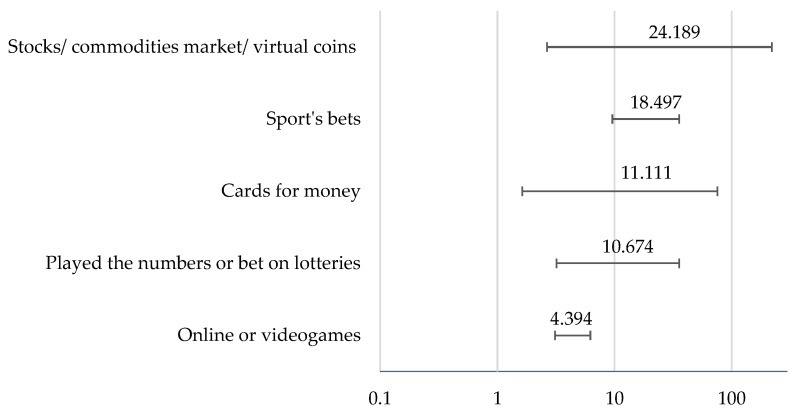
Multivariate analysis of the risk of gambling disorder according to the type of gambling.

**Figure 2 healthcare-11-02527-f002:**
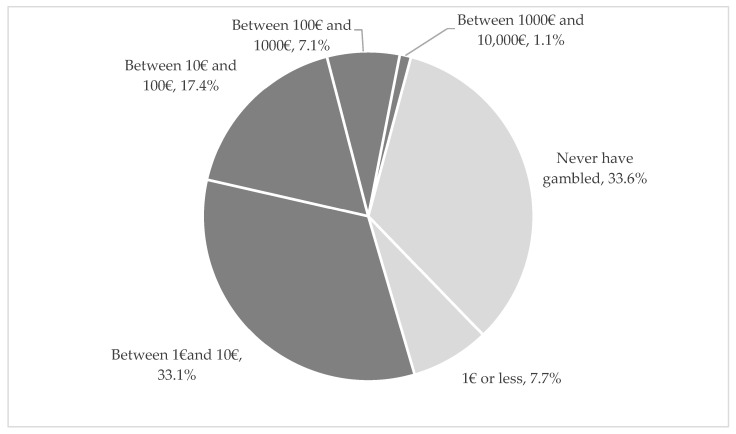
Maximum amount of money gambled in one day (%).

**Table 1 healthcare-11-02527-t001:** Sample characteristics (n = 1123).

Characteristics	n	*%*
Age		
<20 years	407	36.2
[20–22] years	377	33.6
>22 years	339	30.2
Gender		
Female	684	60.9
Male	428	38.1
Other	11	1.0
Marital status		
Single	1061	94.5
Married/Nonmarital partnership	61	5.4
Divorced	1	0.1
Socioeconomic class		
Low	168	15.0
Medium	923	82.2
High	32	2.8
Had to switch accommodations		
Yes	396	35.3
No	727	64.7
Academic year		
First	315	28.0
Second	188	16.7
Third	163	14.5
Fourth	59	5.3
Fifth	55	4.9
Sixth	47	4.2
Master	214	19.1
Postgraduate	2	0.2
Doctorate	80	7.1
Scholarship		
Yes	347	30.9
No	776	69.1
Employed		
Yes	182	16.2
No	941	83.8
Mental health status		
Excellent	45	4.0
Very good	241	21.5
Good	390	34.7
Reasonable	357	31.8
Bad	90	8.0
Physical health status		
Excellent	55	4.9
Very good	323	28.8
Good	468	41.7
Reasonable	235	20.9
Bad	42	3.7
Substance consumption		
Alcohol	555	49.4
Tobacco	157	14.0
Cannabis	98	8.7
Ecstasy	7	0.6
Cocaine	1	0.1
Amphetamines	2	0.2
LSD	3	0.3
Others	13	1.2
None	537	47.8

**Table 2 healthcare-11-02527-t002:** Gambling disorder by SOGS-PV scores.

	SOGS-PV Score	n (%) n = 1123	95% CI
No gambling disorder	0	902 (80.3)	77.9–82.6
Probable pathological gambler (PPG)	[1–4]	186 (16.6)	14.4–18.8
Problem gambler (PG)	≥5	35 (3.1)	2.1–4.1

**Table 3 healthcare-11-02527-t003:** Types and frequency of gambling and risk for gambling disorders.

	Less Than Once per Week n (%)	Once Or More per Week n (%)	Gambling Disorder ^†^ (SOGS ≥ 1) OR [95% CI]	*p*
Online or videogames	270 (24.0)	325 (28.9)	4.243 (3.119–5.772)	<0.001
“Scratch card” games	391 (34.8)	22 (2.0)	5.122 (2.183-12.014)	<0.001
Games of skill for money	385 (34.3)	15 (1.3)	1.493 (0.471–4.734)	0.496
Played the numbers or bet on lotteries	279 (24.8)	17 (1.5)	10.300 (3.590–29.555)	<0.001
Sports bets	129 (11.5)	66 (5.9)	19.516 (10.576–36.013)	<0.001
Casino	101 (9.0)	6 (0.5)	- §	-
Stocks/commodities market/virtual coins *	69 (6.1)	12 (1.1)	47.195 (6.060–367.569)	<0.001
Played bingo	78 (6.9)	3 (0.3)	8.228 (0.743–91.156)	0.086
Cards for money	61 (5.4)	9 (0.8)	14.720 (3.036–71.358)	<0.001
Slot machines	62 (5.5)	6 (0.5)	20.856 (2.424–179.437)	0.006
Dice games for money	6 (0.5)	2 (0.2)	- §	-
Animal bets	4 (0.4)	0 (0.0)	-	-
Other gambling activities **	82 (7.3)	67 (6.0)	1.190 (0.657–2.156)	0.566

† Logistic regression. Compares individuals who engage in gambling activities at a frequency of once per week or more, in contrast to those who either abstain from gambling entirely or engage in it less frequently than once per week. § All the positives present a high risk for gambling disorders. * Includes commodities market/virtual coins market; ** Includes card games without money; mobile games; board games; sports; Russian roulette; role-playing games; and various other types of video games (The Sims; Valorant; Cooking Diary; Homescapes; Idle Miner; Roblox; Minecraft; Plants vs. Zombies).

**Table 4 healthcare-11-02527-t004:** Univariate analysis, adjusted odds ratio for gambling disorder, and significant variables.

Variables	Score 0 n (%)	Score ≥ 1 n (%)	*p* (Univariate Analysis)	OR [95% CI]	*p* (Logistic Regression)
Age	21.0 [17–64]	20.0 [17–48]	0.32	0.953 [0.922–0.986]	0.005
Gender					
Female	605 (54.4%)	79 (7.1%)	< 0.001	Reference	< 0.001
Male	288 (25.9%)	140 (12.6%)		2.756 [1.899–4.000]	
Education area					
Health	304 (27.1%)	42 (3.7%)	< 0.001	Reference	
Arts	12 (1.1%)	0 (0.0%)		-	0.998
Architecture	23 (2.0%)	6 (0.5%)		2.026 [0.680–6.032]	0.205
Social and Behavioural Sciences	51 (4.5%)	10 (0.9%)		1.346 [0.572–3.169]	0.496
Education	72 (6.4%)	12 (1.1%)		1.355 [0.631–2.910]	0.436
Personal services	23 (2.0%)	13 (1.2%)		2.858 [1.179–6.929]	0.020
Engineer	142 (12.6%)	49 (4.4%)		1.252 [0.721–2.174]	0.425
Business	50 (4.5%)	20 (1.8%)		1.350 [0.641–2.846]	0.430
Linguistics	62 (5.5%)	21 (1.9%)		2.984 [1.538–5.788]	0.001
Life science	139 (12.4%)	41 (3.7%)		1.433 [0.825–2.491]	0.201
Law	24 (2.1%)	7 (0.6%)		1.401 [0.487–4.029]	0.531
The largest amount of money ever gambled in one day					
Less than 1€	429 (38.2%)	35 (3.1%)	< 0.001	Reference	< 0.001
More than 1€	473 (42.1%)	186 (16.6%)		3.938 [2.580–6.012]	
Influenced during COVID-19 pandemic					
Yes	81 (7.2%)	101 (9.0%)	< 0.001	Reference	< 0.001
No	821 (73.1%)	120 (10.7%)		0.129 [0.087–0.191]	
Relatives with gambling disorder					
Father	33 (2.9%)	14 (1.2%)	0.075	-	0.116
Mother	4 (0.4%)	5 (0.4%)	0.007	5.284 [1.038–26.902]	0.045
Sibling	18 (1.6%)	8 (0.7%)	0.150	-	0.210
Grandparent	42 (3.7%)	11 (1.0%)	0.840	-	0.865
Partner	8 (0.7%)	2 (0.2%)	0.980	-	0.797
Son or daughter	1 (0.1%)	0 (0.0%)	0.620	-	0.809
Other relatives	105 (9.3%)	18 (1.6%)	0.136	-	0.166
A friend	64 (5.7%)	28 (2.5%)	0.007	-	0.364
Mental health status					
Excellent, good, or very good	548 (48.8%)	128 (11.4%)	0.440	Reference	0.483
Reasonable or bad	354 (31.5%)	93 (8.3%)			
Physical health status					
Excellent, good, or very good	679 (60.5%)	167 (14.9%)	0.929	Reference	0.187
Reasonable or bad	223 (19.9%)	54 (4.8%)			
Substance consumption					
Any substance	451 (50.0%)	135 (61.1%)	0.003	-	0.069
Alcohol consumption	428 (38.1%)	127 (11.3%)	0.008	-	0.140
Tobacco consumption	116 (10.3%)	41 (3.7%)	0.029	-	0.059
Cannabis consumption	67 (6.0%)	31 (2.8%)	0.002	-	0.140
Ecstasy consumption	5 (0.4%)	2 (0.2%)	0.553	-	0.687
Amphetamines consumption	2 (0.2%)	0 (0.0%)	0.484	-	0.584
LSD consumption	3 (0.3%)	0 (0.0%)	0.391	-	0.408
Other substances consumption	8 (0.7%)	5 (0.4%)	0.087	-	0.670

OR: odds ratio; CI: confidence intervals. *p* is considered significant if <0.05. Adjustment variables: gender and education area.

## Data Availability

The data presented in this study are available on request from the corresponding author. The data are not publicly available due to privacy and ethical restrictions.

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
