# Peer review of "Gambling Disorder among Porto’s University Students"

_healthcare, 2023, doi:10.3390/healthcare11182527_

Round 1
Reviewer 1 Report
This paper reports the findings of a cross-sectional survey of Portuguese students, looking at predictors of pathological gambling using the SOGS. The study itself is of interest, the data appear sound, and the methods were well reported. I particularly liked that the survey materials were provided as a supplementary material. However, I do have a number of concerns about the manuscript as it presently stands. There were areas of the results that were unclear, the introduction/discussion could benefit from substantial revisions, and the definition of some of the gambling categories is potentially problematic.
- Both the introduction and discussion would benefit from substantial development. Neither gave me a particularly strong sense of what the paper was about or what it found. The majority of the introduction gives a very general overview of gambling disorder and its predictors. It didn't give a strong sense of why there might be specific aspects or features of interest that necessitate the study (e.g. are there specific features of Portugal's gambling market that might be different, the impact of COVID?). Similarly there were bits of the discussion (e.g. implications) where more specificity would be helpful, and how this research could feed into improvements in screening or effective therapies.
- The point regarding tobacco being a predictor in the abstract and discussion appears to be incorrect. The p value (.059) is not statistically significant.
- The justification for the multivariate regression was not clear in Table 3. A number of effects that were significant in the univariate analysis were not included in the final model, and it was not clear what selection criteria were used to identify which variables were included in the final analysis. Further details of this could be included in the results, or under a statistical methods subheading in the methods. This feeds into the points in the discussion about alcohol and cannabis not being significant - in the univariate analyses, they appear to be (p < .01), but there is no value reported for the regression. It's not clear from what is reported whether they were tested but not significant, or not included in the regression analysis.
- The figures could really benefit from some revision - at the moment they do not look publication quality. Especially Figure 2. I would recommend replacing with a bar chart or a table. The use of multiple columns within Figure 1 makes it harder to understand the findings - it might be preferable to have two sets of bars, one for past year engagement at all, and a second for past week or more frequent engagement.
- I was surprised to see there wasn't an analysis testing whether specific types of gambling game were associated with higher SOGS scores or PG prevalence. I think that this could be included in Table 3. Similarly, the number of gambling games played could also be included in the first column of Table 3, given the literature on gambling involvement and pathological gambling.
- The categorization of different types of gambling is highly questionable. The authors have combined online gambling and videogaming into a single category in Figure 1. The legend has a number of different asterisks, but it is not clear which these refer to. As such, I'm concerned that this may overstate the risks of online gambling due to the choice of defined categories. Did the authors specify specific types of games (e.g. ones with loot boxes?). In either event, this means the category of online gambling is much broader than how it is traditionally conceptualized in the gambling literature.
- I would take the findings regarding changes in gambling during the COVID pandemic with a couple of qualifications. First, as the changes are self-reported, this might reflect biases related to gambling disorder (e.g. preoccupation) rather than the other way around. Second, as a young sample, this period overlaps with participants' reaching the legal gambling age, which I believe is 18 in Portugal. Thus increases in gambling during COVID may be confounded with legally being allowed to gamble.
English language is generally fine. In terms of grammar, the introduction had a lot of short, single sentence paragraphs that feed into my comments regarding developing the introduction, and detract from the overall flow of the manuscript.
Reviewer 2 Report
Why gambling disorder is an issue needs to be covered in the study. this will demonstrate the issue that led to the study's being carried out. why is gambling disorder a problem globally and also in Portugal. why did the study chose university students? what's so special about them.
The study claims that little has ben done in this study area. This needs to be proved. There is a need for the study to show that a comprehensive literature review was done. You can add a paragraph in the background/introduction section to capture this.
Materials and methods
It would be better if you structure/sub-divide this section into different sections i.e data collection instrument, population and sample, estimation techniques, data analysis etc. This will better present the section and it will also show the logical sequence that the data gathering and analysis process followed.
Reviewer 3 Report
Dear Authors,
Congratulations on this prevalence study of gambling among university students in Portugal.
I have two somewhat substantial suggestions and a couple of minor things.
* You state on Page 11 lines 294-6 that the results are generalisable because you obtained a large sample size, and surveys were anonymous and confidential. These are not criteria for generalisabilty of prevalence estimates. Please correct this statement.
* Can you provide a brief discussion of the value of weighting and why it could not be performed on your data?
Minor changes:
* Delete 'evaluation' from Line 47 - it's not needed.
* Line 65-66 "can lead to change in a person's life" does not really express the hardship, distress and devastation that gambling can cause. It is also very vague language. I suggest rephrasing.
All the best with the paper.
Reviewer 4 Report
Please see comments in the attached file

Aside from the comment about use of non-gender-biased pronouns and adjectives (see attached) the English was very good.
Reviewer 5 Report
The authors present a cross-sectional study that evaluates the prevalence of gambling disorder among university students from the city of Porto in the North of Portugal. 1123 subjects participate in the study (61% self-identified as women). The majority of participants (66,4%) are engaged in gambling activities and 19.7% could have a gambling condition (3,1% with probable gambling disorder). Males, tobacco users, and having a mother with gambling problems were identified as risk factors. They also identified changes in gambling habits after covid as a contributing factor.
The work could contribute to the knowledge on pathological gambling among university students, to raise awareness on this problem, and to develop preventive effective measures to reduce its burden. There are some issues that need to be clarified.
A. The first paragraph seems to be a mere description of the DSM-5 criteria. The authors should rewrite the definition of the disorder. If needed, they could present the DSM-5 criteria as a table.
B. The Portuguese context was not sufficiently presented. Previous publications on gambling in Portugal were not included in the contextualization of the study. There are studies from the same institution that were ignored by the authors… Why is pathological gambling a relevant topic in Portugal? Are there ongoing discussions?
C. The Materials and Methods section should be divided into subsections: study design, participants, instruments, ethical considerations, and statistical analysis.
D. There are some methodological problems with the questionnaire: evaluating the economic status by asking a qualitative group is not a good option; be careful when discussing it.
E. The questionnaire asks about “substance use” but in the paper, it is mentioned as “substance abuse”.
F. The question about the pandemic and gambling habits is not as informative as it looks. Considering the age of the participants, it seems that the majority of participants began their university studies during the pandemic. It could be interpreted as a more significant factor than the pandemic itself…
G. Did you include all the variables on the binary logistic regression? Please clarify. No results were found for Mental health status or physical health status.
H. The study identified “online or video games” as the most popular activity. Is this gambling or gaming? How do you know that gambling is involved in online or video games? Please clarify or, at least, discuss this issue in the discussion section.
I. Scratch card games seem to be more popular among university students than in other prevalence studies. Is this an issue in Portugal?
J. Sports betting seems to be highly popular among university students. Please comment on it considering the Portuguese context, advertising in soccer matches, and Portuguese law.
K. Please highlight the primary prevention in your discussion section providing suggestions on specific and contextualizing measures that could prevent these phenomena.
Ok.
Round 2
Reviewer 1 Report
I appreciate the efforts the authors have gone to address the comments from the first round of review. Nonetheless, I have a number of major concerns that have not been fully addressed. In particular, I am concerned about a series of reporting errors in the regression.
- I don't think excluding non-significant results from a multivariable model is appropriate practice. Several variables entered are non-significant (i.e. different levels of education area), conflicting with the passage entered regarding the exclusion of non-significant variables. I would recommend including the findings for all of the variables entered, as they are being modelled simultaneously. Not reporting the non-significant results means that the model is not being fully reported, which is not ideal statistical practice.
- The data around tobacco still concerns me. The authors in their reply noted that the .059 was a typo, and should be .029. However, the CI's reported on the odds ratio are .982 - 2.517. This would therefore suggest the p value should be non-significant, as the confidence intervals bisect 1. Indeed, I went back and calculated the p value from the OR, and CIs using the formulae from https://www.bmj.com/content/343/bmj.d2304, and came to a p value of .0589 (indicating the original p value of .059 is correct). Could the authors have a second look at this and see what's going on?
- Similarly, there are a couple of issues with the odds ratios reported elsewhere in Table 4. The male OR does not have CI's reported. Also, for personal services, the an OR of 2.936 with CIs of 1.208 and 7.136 is reported, with a p value of .17. I would recommend the authors have a look at this, as again calculating the p value suggests this is significant (.017).
- I appreciate the authors have explained the distinction in their online gambling - videogames category, but I'm still concerned about the logic of the distinction between online and videogames, and also the other gambling activities (which includes "various other types of video games (The Sims; Valorant; Cooking diary; Homescapes; Idle Miner, Roblox; Minecraft; Plants vs Zombies"). The authors reply notes that the open option is chosen if people didn't endorse other gambling activities. If anything, this reinforces my concern that there are individual differences in how respondents perceive these activities, and the approach taken introduces a lot of overlap between online gambling, gaming with loot boxes, and gaming in general. I think this needs to discussed as a limitation of the present study in the discussion. Although there is a literature on the overlap between loot boxes and gambling, it is really atypical to treat these activities in the way they have been done in this study.
English language is fine
Reviewer 4 Report
The authors have answered most of my questions. With respect to the issue of ‘representativeness’ my question was intended to invite the authors to comment on similarities and differences between their sample and findings from other postsecondary / university student samples that have completed the SOGS. i.e., Is there something unique about Porto or is the pattern seen here much the same as in other student samples?
See the following relevant sources: https://pubmed.ncbi.nlm.nih.gov/?term=36612781%2C26209271%2C32728984%2C26009785 %2C32762419%2C35133668&filter=years.2013-2023&size=100
Sex/gender reliably affect etiology and profile of GD https://pubmed.ncbi.nlm.nih.gov/?term=26935871%2C36612781%2C34661803%2C29105942 %2C23183847%2C27260007%2C37115422&size=100
As such, it would be useful to note in Limitations section, that the relatively modest N/gender in this sample precluded gender-based regression analysis, which may have revealed important interactions with other predictors
Reviewer 5 Report
Congratulations. The paper was really improved after revision.
Author Response
I wanted to to express my sincere gratitude for the feedback and the thoughtful review you provided for our manuscript.
Please feel free to reach out if you have any further comments, suggestions, or insights. Your expertise is highly respected and deeply appreciated.
Thank you once again for your time and dedication.